# The Epstein-Barr Virus Hacks Immune Checkpoints: Evidence and Consequences for Lymphoproliferative Disorders and Cancers

**DOI:** 10.3390/biom12030397

**Published:** 2022-03-04

**Authors:** Alison Felipe Bordini Biggi, Deilson Elgui de Oliveira

**Affiliations:** 1Biosciences Institute of Botucatu, São Paulo State University (UNESP), Botucatu 18618-689, SP, Brazil; felipe.biggi@unesp.br; 2Department of Pathology, Medical School, São Paulo State University (UNESP), Botucatu 18618-687, SP, Brazil; 3ViriCan, Institute for Biotechnology (IBTEC), São Paulo State University (UNESP), Botucatu 18607-440, SP, Brazil

**Keywords:** Epstein-Barr virus, immune checkpoints, Hodgkin lymphoma, non-Hodgkin lymphomas, gastric adenocarcinomas, EBV-associated, viral oncoproteins, viral ncRNAs, immunotherapy, immune checkpoint blockade, cancer management

## Abstract

The Epstein-Barr Virus (EBV) is a gammaherpesvirus involved in the etiopathogenesis of a variety of human cancers, mostly of lymphoid and epithelial origin. The EBV infection participates in both cell transformation and tumor progression, also playing an important role in subverting immune responses against cancers. The homeostasis of the immune system is tightly regulated by inhibitory mechanisms affecting key immune effectors, such as T lymphocytes and NK cells. Collectively known as immune checkpoints, these mechanisms rely on a set of cellular receptors and ligands. These molecules may be candidate targets for immune checkpoints blockade—an emergent and promising modality of immunotherapy already proven to be valuable for a variety of human cancers. The EBV was lately suspected to interfere with the expression of immune checkpoint molecules, notably PD-1 and its ligands, found to be overexpressed in cases of Hodgkin lymphoma, nasopharyngeal, and gastric adenocarcinomas associated with the viral infection. Even though there is compelling evidence showing that the EBV interferes with other immune checkpoint regulators (e.g., CTLA-4, LAG-3, TIM-3, and VISTA), the published data are still scarce. Herein, we discuss the current state of the knowledge on how the EBV interferes with the activity of immune checkpoints regulators, as well as its implications considering the immune checkpoints blockade for clinical management of the EBV-associated malignancies, notably lymphomas.

## 1. Introduction

The World Health Organization (WHO) estimated that there were more than 19 million new cancer cases worldwide for 2020, and almost 10 million associated deaths [1]. Wealthy countries are likely to have higher cancer incidences, but the impact of these diseases in developing countries is even more significant [2]. Excluding non-melanoma skin cancers, a conservative estimate of 2.2 million cases of cancers were attributable to infections in 2018—about 13% of cancer incidence worldwide [3]. Viral infections by the HPV, the hepatitis viruses B (HBV) and C (HCV), the human T-lymphotropic virus type 1 (HTLV-1), the Kaposi sarcoma herpesvirus (KSHV), and the Epstein-Barr virus (EBV) are recognized as human carcinogens [4].

Formally designated Human Gamma Herpesvirus type 4 (HHV-4), the EBV deserves special attention because it persistently infects most human adults worldwide. The virus is transmitted by saliva and the contagion starts with infection of epithelial cells in the oropharynx, where the EBV replicates and reaches local B cells to establish the latent infection [5,6]. Only a small fraction of people infected for life will develop the EBV-associated cancer. Nonetheless, it is well established that the EBV participates in the etiopathogenesis of a variety of human cancers, notably the endemic forms of Burkitt lymphomas (BL) and nasopharyngeal carcinomas (NPC). The virus is also readily detected within the neoplastic cells of a variable number of cases in other malignancies, notably classic Hodgkin lymphomas (cHL), non-Hodgkin’s lymphomas (NHL), and a small fraction of gastric adenocarcinomas (GaC) [7].

Most of the neoplastic cells are latently infected in the EBV-associated cancers, expressing only a subset of viral products. According to the repertoire of latent viral products expressed, three main EBV latency programs are typically described, known as viral latency types I, II, and III. All the EBV latent products are expressed in latency type III, including six nuclear EBV antigens (EBNAs 1, 2, 3A, 3B, 3C, and LP), three latent membrane proteins (LMPs 1, 2A, and 2B), and viral non-coding RNAs (ncRNAs), such as the EBV-encoded small RNAs (EBERs 1 and 2) and miR-BARTs. Latency type II is more restricted, featuring the expression of the EBV oncoproteins EBNA1, LMP1, and LMP2A, while most of the viral products are repressed in latency type I, characterized by the expression of EBNA1 and EBERs only. These viral latency expression profiles are observed in latently infected non-neoplastic cells in vivo and they are also recapitulated in EBV-infected malignant cells in cancers [5].

The EBV is also remarkable in how it exploits its hosts’ immune systems. For instance, the activity of several components of the interferon response, such as IFNα4 and IFNβ, is inhibited by the viral immediate-early gene BZLF1 [8]. Additionally, the viral DNase/alkaline exonuclease BGLF5 downregulates Toll-like receptors (TLR) [9,10], impacting negatively on the recognition of pathogen-associated molecular patterns (PAMPs). The elimination of the EBV-infected B lymphocytes by NK cells is adversely affected by the expression of a viral homolog of the human immunosuppressive interleukin 10 (vIL-10) [11]. The EBV infection also compromises immune responses mediated by the Major Histocompatibility Complex (MHC) molecules: for instance, the viral protein BZLF1 post-translationally downregulates CD74, involved in antigen presentation mediated by MHC class II molecules [12], while the EBV BGLF5 induces shutoff of the cellular protein synthesis machinery, affecting adversely the MHC-I-mediated recognition of infected cells by CD8^+^ T lymphocytes [13]. Both the EBV proteins vIL-10 and BNLF2a were also found to compromise the presentation of antigens by infected cells in the context of MHC-I molecules [11]. Therefore, both innate and immune responses are manipulated by the EBV, which explains how successful this virus is establishing lifelong persistent infection in humans.

Both the EBV proteins [14] and viral ncRNAs [15] have known oncogenic properties in human cells. Because the immune evasion is an important feature of cancers in general [16], the modulation of immune responses by the EBV proteins (e.g., LMP1 [17]) and viral ncRNAs [18,19] can be instrumental for virus-induced carcinogenesis. Expanding our understanding of how the EBV infection hacks the immune system by exploiting the immune checkpoints can sheds new light on the mechanisms of viral carcinogenesis and cancer biology, providing new and important opportunities for better management of these potentially lethal diseases. Therefore, herein, we summarize the state of the knowledge on this topic, aiming to highlight the impact of the immune checkpoints regulation by viral products in the EBV-associated cancers, as well as the consequences considering new therapies based on immune checkpoint blockade. For more details about the interplay between the EBV and immune regulatory molecules, we recommend a recent review by Münz (2021) [20].

Because the regulation of immune checkpoints is especially relevant for cancers of lymphoid origin, we will focus on the EBV-associated lymphomas, overviewed in the following section. Nonetheless, other EBV-associated cancers will be also briefly commented on throughout this review.

## 2. Overview of the EBV-Associated Lymphomas

Lymphomas comprise a broad group of cancers derived from transformed lymphoid cells that are conventionally divided as Hodgkin (HL) and non-Hodgkin lymphomas (NHL) in humans. The former is a more well-characterized entity, while the NHL group is notable for its diversity. There are remarkable different diseases among NHL, such as the highly aggressive BL, the often-indolent follicular lymphoma (FL), and the very heterogeneous group of diffuse large B cell lymphomas (DLBCL). These are phenotypically and clinically mature lymphoid B-cell malignant neoplasms [21], and they are mostly derived from the GC—structures of the secondary lymphoid tissues in which the selection of B cells producing high-affinity antibodies against foreign antigens occurs. Within the GC, B cells undergo somatic mutation of genes encoding immunoglobulin chains, which constitute the B cell receptor (BCR) expressed on the surface of B-cells. Known as GC response, this process is tightly regulated for the survival of only B-cell clones capable of effective responses against foreign antigens, eliminating subpopulations of self-reactive B cells. The GC structure and the GC response homeostasis rely on the coordination of multiple cell types other than B-cells, such as stromal cells, follicular T helper (Th) cells, and follicular T regulatory (Tr) Foxp3+ cells [22]. Of note, the EBV infection regulates many aspects of B-cell ontogenesis, allowing the emergence of different B-cell lymphomas according to the maturation stage of the B-cell and its status regarding the GC response [23,24].

The HL is divided into the classical form (cHL), most cases, and the rare nodular lymphocyte-predominant form (NLPHL). The tumor has a scarce population of giant neoplastic cells, referred to as Hodgkin or Reed Sternberg (HRS) cells, surrounded by a variable and heterogeneous repertoire of reactive non-neoplastic cells [25]. The EBV is detected in HRS cells in roughly half of all cHL cases, and in all cHL arising in people living with infection by the human immunodeficiency virus (HIV) [26]; conversely, it is typically absent in NLPHL, except for a few reports [27,28]. The HRS cells have acquired somatic mutations in the variable (V) segment of the IGH gene, which is indicative that the HL clonally arises from GC or post-GC B-cells [29,30].

The BL is the most common NHL associated with the EBV, and it has three epidemiological forms: the endemic (eBL), sporadic (sBL), and immunosuppression-associated BL (iBL), often arising in people living with HIV (PLHIV). The eBL is frequently observed in the malaria belt region in Africa and Papua New Guinea and virtually all cases are EBV-associated. Non-endemic BLs (sBL and iBL) are rare worldwide, and the EBV is associated with these forms in only a subset of the cases [21]. All BL cases share a pathogenetic hallmark of recurring translocations involving immunoglobulin heavy chain genes (IGH) and *MYC,* which was the first proto-oncogene linked to GC-derived lymphomagenesis [31].

Besides BL, other B-cell NHL associated with EBV infection include the EBV-positive DLBCL not otherwise specified (NOS) [32], as well as lymphomas more frequently observed in PLHIV [33]. Because HIV hinders the immune system, the EBV-infected cells in PLHIV are less vulnerable to the immune defenses, favoring the emergence and expansion of EBV-induced malignant cells, consequently increasing the risk for both cHL and NHL. Worth noting, the HIV/EBV co-infection in humans foster both the HIV disease progression and the lymphomagenesis induced by EBV [34]. The EBV is also associated with lymphoproliferative disorders and NK/T lymphomas (NKTL), but the viral contribution to the development of these diseases still is elusive [35].

The impact of the EBV infection on the clinical behavior of lymphomas can be appreciated in terms of reduced overall survival (OS) and progression-free survival (PFS) in EBV-associated HL [36,37] and DLBCLs [38,39], for instance. These data stress the need for more effective therapies against EBV-associated lymphomas. Recently, treatments based on immune checkpoint blockade emerged with great promise for the management of human cancers, but the number of patients achieving complete responses with this new therapeutic modality remains to be improved. Thus, understanding how the EBV regulates immune checkpoints poses a strategic role for cancer control—not to mention its putative prognostic and predictive values for cancers associated with the EBV infection, which will be discussed in the following sections.

## 3. Interplay between Immune Checkpoints and the EBV

The normal immune homeostasis and the activation of T-cells, more specifically, rely on intracellular events triggered by the activation of the T-cell receptor (TCR), upon the MHC-mediated recognition of antigenic peptides. Essentially, normal regulators of the immune checkpoint can modulate T-cell activation by inhibiting TCR co-stimulatory signals. As long known, T-cells become hyporesponsive without co-stimulatory signals provided by some surface receptors, such as CD28 [40,41]. Inhibitory signals triggered by the programmed death receptor 1 (PD-1), and the cytotoxic T-lymphocyte protein 4 (CTLA-4) also control the intensity of T-cell response, which is important to mitigate autoimmunity and allowing peripheral tolerance. This is a homeostatic mechanism important to prevent tissue damage during inflammation, infections, and even neoplastic processes [42].

Currently the literature has more data on some immune regulatory molecules, such as PD-1 and its ligands and CTLA-4, compared to others. Rather than a reliable proxy of the relevance of these molecules in the context of immune checkpoints, this may represent a bias based on the seminal publications about these molecules and their role in the immune system. Furthermore, some immune checkpoints regulators also received more attention from the scientific community after being successfully targeted for immune checkpoint blockade, aiming the development of new protocols in this emerging category of treatment for the management of human cancers.

The PD-1 receptor has two ligands, PD-L1 and PD-L2, which are typically expressed by hematopoietic and epithelial cells. The interactions between PD-1/PD-L1 or CD80/PD-L1 [43] inhibit T-cell response, leading to cellular exhaustion or even apoptosis, which is often observed in chronic inflammation and cancers [44]. On the other hand, CTLA-4 negatively controls T-cell activation by competing with the CD28 co-stimulatory receptor for its ligands CD80 (B7.1) and CD86 (B7.2). Because CTLA-4 has a higher affinity for CD80 compared to CD28, it antagonizes the CD28-mediated co-stimulation, reducing or inhibiting the T-cell activity [45,46]. The PD-1 and CTLA-4 receptors behave differently regarding the inhibition of T-cell activation and responses: PD-1 acts on differentiated T-cells effectors, while the activity of CTLA-4 affects mostly the T-cell priming. Additionally, PD-1 ligands (PD-Ls) are broadly expressed, playing a role in peripheral T-cell tolerance against adverse immune reactions; conversely, CTLA-4 ligands are often expressed by antigen-presenting cells (APCs) on lymph nodes and the spleen [47,48]. Because of its broader activity and considering its upregulation in a variety of human cancers, it is not surprising that more data has been published on the activity of PD-1 and its ligands; consequently, we have a better understanding of the role of PD-1 and its ligands as immune checkpoints regulators compared to other molecules in the same category.

The genes that encode both PD-1 ligands, PD-L1 (*CD274*; HGNC:17635) and PD-L2 (*PDCD1LG2*; HGNC:18731), are located in 9p24.1. This is also the genomic region for the human gene encoding the Janus kinase 2 (*JAK2*; HGNC:6192)—a key component of the JAK/STAT pathway, involved in intracellular signaling following cytokine and growth factors stimuli. A selective amplification of 9p24.1 leading to an increase in the expression of PD-Ls was reported in cHL and other hematologic malignancies, including the primary mediastinal large B-cell lymphoma (MLBCL), a specific DLBCL subtype [49]. In 2012, Green and colleagues reported that 9p24.1 was amplified in 5/12 (41.7%) EBV-negative cHL, but none of the three EBV-positive specimens were evaluated [50]. This result might suggest that 9p24.1 amplification and EBV infection are mutually exclusive mechanisms leading to upregulation of PD-Ls. Later (2016), in a cohort of 118 cHL cases, the frequency of genetic structural alterations involving the PD-L1 and PD-L2 genes (e.g., polysomy, gene copy gains, and translocations) was reported at similar rates comparing EBV-positive and EBV-negative tumors [51]. Therefore, the upregulation of PD-Ls in cHL may be due to genetic alterations involving 9p24.1 (e.g., by causing changes in the copy number of PD-L1/PD-L2/JAK2), to EBV infection, or even a combined result of both events.

The association between the EBV infection and the expression of PD-Ls were previously investigated using a humanized mouse model of DLBCL: while mock-infected B-cells had low levels of PD-L1 and low to no expression of PD-L2 (assessed by flow cytometry), both PD-L1 and PD-L2 had increased expression in the same cells when infected by EBV viral strains B95.8 and M81. Interestingly, the authors identified that some neoplastic cells showed EBV EBNA2 and PD-L1 co-expressed by immunohistochemistry (IHC); remarkably, T-cells from the EBV-positive lymphoma model expressed PD-1 and CTLA-4 on their surface, and antibody blockade of these receptors caused a substantial increase of EBV-specific T-cell responses, reducing the lymphoma burden and increasing the survival of mice [52]. This result is consistent with the T-cell mediated control of lymphoma being impaired by the interaction of PD-1 and CTLA-4 with their ligands produced by the EBV-infected B-cells, mostly neoplastic ones.

The association of the EBV infection, upregulation of PD-1/PD-Ls, and lymphomagenesis demonstrate that this virus has the necessary tools to interfere with the immune checkpoint machinery. On the other hand, it should be noted that genomic abnormalities involving mostly the 3′UTR of PD-L1/PD-L2 (e.g., in tandem duplication, inversion, translocation, and deletions) were described in EBV-associated lymphoproliferative disorders of T and NK cells, including the extra-nodal NK T-cell lymphoma (NKTL) and the aggressive NK/T-cell leukemia. Based on a TCGA dataset of more than 10 thousand tumor samples (33 different cancer types), a higher frequency of PD-L1 and PD-L2 structural variations was observed in EBV-associated DLBCL, peripheral T-cell lymphomas, and gastric adenocarcinoma, compared to cases without viral infection. Interestingly, PD-L1 variations were found in T/NK-cells lymphomas, while PD-L2 variations were more restricted to B-cell lymphomas and the B-cell lineage [53]. Whether an EBV infection has any direct or indirect causal role in this high frequency of genomic changes that culminate in increased PD-Ls expression remains to be uncovered. Nevertheless, some aspects of mechanisms driven by an EBV infection for the modulation of immune checkpoints can already be appreciated, as discussed next.

### EBV Regulation of Immune Checkpoints: Insights Based on PD-1/PD-Ls Upregulation

In cells latently infected by the EBV showing latency types II (neoplastic cells of cHL and NPC, for instance) or III (e.g., EBV-immortalized lymphoblastoid cell lines, PTLD, and lymphomas in immunocompromised hosts), upregulation of PD-L1 can be a consequence of LMP1 expression. This viral oncoprotein can increase the activity of the PD-L1 promoter by stimulating the signaling pathways mediated by STAT5 and JAK3—as reported in the EBV-transformed lymphoblastoid cell line NOR-LCL, in which the PD-L1 gene enhancer is active and responsive to the cJun/AP-1 transcription factors induced by LMP-1 [50]. Comparing the cell lines SNK-6 (RRID: CVCL_A673) and NK-92 (RRID: CVCL_2142), both derived from NK/T cell lymphoid malignancies, PD-L1 expression was significantly higher in the former, which is EBV-positive, compared to the EBV-negative NK-92. Furthermore, the EBV-negative NK-92 cells showed an increase in PD-L1 protein after lentiviral transduction to express EBV LMP1, effects attributed to the LMP-1-induced activation of the NF-kB and MAPK signaling pathways [54].

Similar results also implicate LMP1 in PD-L1 upregulation in the EBV-associated epithelial cancers. For instance, PD-L1 is expressed at remarkably higher levels in the EBV-positive NPC cell line C666-1 (RRID: CVCL_7949) compared to the EBV-negative cell line NPC-TW03 (RRID: CVCL_6010), or the pre-malignant nasopharyngeal cells NP69^SV40T^ (RRID: CVCL_F755). The increased production of PD-L1 was observed either by infecting NPC-TW03 with the EBV or transfecting this cell line with a recombinant vector for LMP1 expression. Nonetheless, PD-L1 expression was knocked down in LMP-1-transfected NPC-TW03 cells treated with a specific siRNA targeting LMP1 transcripts. This study concluded that the upregulation of PD-L1 by LMP1 was attributed to activation of the signaling pathways NF-κB, JAK3/STAT3, and MAPK/AP1 [55]. Worth noting, because some cell lines used in this study were not reliable models for nasopharyngeal carcinomas (5-8F, 6-10B, CNE-1, CNE-2, HNE-1, and SUNE-1) [56,57,58], some of the states in this study must be interpreted with caution.

Another EBV protein implicated in PD-L1 upregulation is EBNA2, notably in BL and DLBCL. Consistent upregulation of the PD-L1 protein after in vitro infection with a recombinant EBV Akata strain was associated with the EBNA2 expression (but not LMP1) in the EBV-negative BL cell lines OMA4 and DG-75 (RRID: CVCL_0244), as well as the GC-type DLBCL cell lines U-2932 (RRID: CVCL_1896) and SU-DHL-5 (RRID: CVCL_1735). The same study identified that EBV EBNA-2 interacts with the cellular early B-cell factor 1 (EBF1) to repress the expression of mir-34a, which inhibits the PD-L1 transcript, thus increasing the PD-L1 levels. Furthermore, in a cohort of 27 DLBCL, the authors found that the number of PD-L1 positive cells was substantially higher in the EBV-positive tumors compared with the EBV-negative ones and it was also increased significantly in the EBV-positive DLBCL cases expressing EBNA2, compared to cases lacking EBNA2 immunostaining [59]. As LMP1 is expressed in viral latency types II and III, and EBNA-2 is typically restricted to latency type III, these data indicate that PD-L1 upregulation can be exerted by latent proteins expressed during distinct EBV latency programs.

Besides known viral oncoproteins such as LMPs and EBNAs, EBV non-coding RNAs (ncRNAs) are also being considered relevant factors in EBV-induced carcinogenesis [15]. Considering the EBV-driven mechanisms of PD-Ls upregulation, the 3′-UTRs of both PD-L1 and PD-L2 were found to be targeted by the viral miR-BHRF1-2-5p, as experimentally confirmed by luciferase reporter assays. Lymphoblastoid cell lines (LCL) infected with the EBV strains WIL (EBV-WIL) or B95-8 (EBV-B95-8) showed a significant reduction in PD-L1 and PD-L2 when treated with miR-BHRF1-2, indicating that this viral miR-RNA regulates the PD-Ls [60]. In both in vitro cell models and tumor biopsies of NPC and GaC, PD-L1 upregulation was found to be caused by EBV microRNAs (miRs) BARTs 11 and 17-3p, via downregulation of FOXP1 and PBRM1 transcripts, respectively [61]. Additionally, in epithelial malignant cells, the ectopic expression of the EBV miR-BART5-5p in the EBV-negative GaC cells SNU-601 (RRID: CVCL_0101) downregulated the negative STAT3 regulator PIAS3; conversely, treating EBV-positive GaC cells YCCEL1 (RRID: CVCL_L440) with anti-miR-BART5-5p decreased the levels of activated STAT3 (pSTAT3) along with PD-L1. Furthermore, based on a cohort of 103 cases of EBV-associated GaC, the same study reported that 50% of the GaC tumors expressed PD-L1, and those patients had significantly lower OS compared with patients with EBV-associated GaC tumors lacking PD-L1 expression [62].

In summary, the induction of PD-Ls in the EBV-associated cancers (either of lymphoid or epithelial origin) may rely on the activity of both viral proteins and ncRNAs. It should be noted that specificities in the mechanisms of PD-Ls induction by the EBV may be due to pathogenetic features of specific diseases. Nonetheless, the upregulation of PD-Ls and other immune checkpoint regulators in the EBV-associated cancers may have important prognostic consequences, as further discussed in the following section.

## 4. Clinical Impact of Immune Checkpoints in Human Cancers

By interrogating host-virus interactions with a transcriptome analysis with various cancers, Chakravorty et al. (2019) found that the interferon (IFN)-activation status signature discriminates the EBV-associated diseases as IFN-activated (IFN^+^) cancers, such as GaC, and NPC; and IFN-inhibited (IFN^−^) cancers, comprising BL (either endemic or sporadic), angioimmunoblastic T-cell lymphoma (AITL), and NKTCL. The EBV-associated cancers classified as IFN^+^ feature upregulation of PD-L1 and other immune checkpoint regulators, such as the indoleamine 2,3-dioxygenase (IDO)-1 enzyme. Remarkably, these markers strongly distinguished between the EBV-associated and the EBV-negative GaC. Moreover, the treatment of GaC-derived cell lines with exogenous IFN-gamma upregulated PD-L1 and derepressed the expression of IDO1 in the EBV-positive cell line SNU-719 (RRID: CVCL_5086), while only increased PD-L1 in the EBV-negative cell line SNU1 (RRID: CVCL_0099) [63].

Both clinical and experimental evidence link the EBV infection to changes in the levels of immune checkpoints regulators. In a series of 433 DLBCL cases, the EBV-positive tumors (*n* = 30; 6.9%) showed significantly higher levels of PD-L1, PD-L2, LAG3, and TIM3 transcripts [39]. In another study with 260 aggressive DLBCL cases, the group of EBV-associated tumors (*n* = 16; 6.2%) more frequently presented B7-H4 overexpression compared to the EBV-negative cases (*n* = 244; 93.8%); furthermore, the EBV-associated DLBC showed a worse prognosis for the disease, either on terms of OS or PFS. Experimentally, upon infection of the EBV-negative GC-type DLBCL cell line Pfeiffer (RRID: CVCL_3326) with the EBV strain B95.8, the authors observed upregulation of B7-H4, and the EBV-positive Pfeiffer cells undergo apoptosis when treated with anti- B7-H4 antibody [64]. In a study of 464 FFPE samples of adenocarcinoma of the stomach, 450 were analyzed for their EBV status, being 430 EBV-negative and 20 EBV-associated GaC. The VISTA expression on immune cells was more often expressed in EBV-positive samples (*n* = 18; 90%), and samples with high PD-L1 expression. On the other hand, the expression of VISTA on tumor cells was associated with PD-L1 expression, but not with the EBV status (0/20) [65].

The field of immune checkpoints in oncology has evolved at a very fast pace in the last decades—a consequence of its high value both on the basic science front and for the clinical management of cancers. As previously mentioned, most of the studies focused especially on PD-1 and CTLA-4. Nonetheless, other immune checkpoint regulators are getting more attention lately, such as LAG-3, TIM-3, and VISTA. The LAG-3 is a receptor with homology to CD4, that binds to MHC II, therefore acting as a negative regulator of CD4+ and CD8+ T-cells. It is commonly expressed on activated T, B, and NK cells [66]. The regulator TIM-3 interacts with Galectin-9 and is constitutively expressed on innate immune cells (e.g., monocytes, DCs, and mature NK cells), and on activated and exhausted T-cells. Antibodies against TIM-3 have been shown to enhance T-cell response [67]. VISTA is commonly expressed on myeloid, T, and NK cells, and has an inhibitory role in immunity due to its T-cell suppressive function [68].

The tumoral expression of these molecules and their receptors are being frequently associated with a worse prognosis in cancers, making it evident that the manipulation of immune checkpoints is a common mechanism for immune subversion in increased aggressiveness during cancer progression. As previously mentioned, high PD-L1 expression was reported to adversely affect the prognosis of the EBV-associated NPC [55] and GaC [62], for instance. Thus, therapeutic strategies to block immune checkpoints have the potential to support the management of various human cancers, especially lymphomas [69].

In a large cohort with 1253 DLBCL cases, the expression of PD-L1 was associated with the non-GC tumor subtype and EBV positivity. Clinical information from 273 patients showed that death due to any reason (disease progression, notably) was more frequent in patients with PD-L1-positive tumors, compared with those lacking PD-L1 expression (59% vs. 32%, respectively), and PD-L1+ cases showed a significantly worse overall survival (OS) [70]. In another series of DLBCL cases, high levels of soluble PD-L1 also showed a similar profile of inferior OS [71]. A meta-analysis with nine studies with 2005 NHL patients also strengthened the perception that PD-L1 expression is associated with worse OS [72], and similar results for DLBCLs patients are also reported for LAG-3 [73].

In a cohort of 514 consecutively resected gastric cancers, the subset of EBV-positive GaC cases (*n* = 32; 6.2%) showed higher PD-L1 expression (overall, in tumor cells, and immune cells) and worse prognosis compared to conventional, EBV-negative GaC cases. Interestingly, five different GaC-derived cell lines evaluated were negatively affected in terms of cell proliferation, cell migration, and invasion in vitro after knocking down PD-L1 with a specific siRNA, and this effect was more pronounced for EBV-positive GaC cells (YCCEL1 and SNU-719) compared to the EBV-negative cell lines (SNU-601, SNU-216, and AGS) [74]. The impact of EBV infection and PD-L1 expression in GaC is also appreciated in another series of EBV-positive cases (*n* = 43), in which formalin-fixed paraffin-embedded (FFPE) tumor specimens were subjected to the estimation of the EBV copy number (EBV-CN) by qPCR. No correlation was observed for EBV-CN and the clinicopathological features of the disease; nevertheless, higher EBV-CN correlated significantly with PD-L1 expression in tumors (as assessed by IHC), and higher EBV-CN were more frequent in cases with PD-L1-expressing tumors, which showed worse disease-specific survival (DSS) compared to cases with low EBV-CN [75].

Based on these data, it is plausible to interrogate whether other EBV-positive cancers show a similar prognostic profile due to the expression levels of immune checkpoint regulators, and to what extent molecules other than PD-1/PD-Ls also play a role in the progression of EBV-associated cancers. Furthermore, elucidating the role of different immune checkpoint regulators opens a new window of opportunity regarding the use of immune checkpoint blockade strategies for the treatment of cancers, as discussed below.

## 5. Immune Checkpoint Blockade in the EBV-Associated Cancers

To date, only PD-1/PD-L1 and CTLA-4 have immune checkpoint inhibitors (ICIs) approved for therapeutic use [76]. Nonetheless, ongoing early-phase clinical trials are targeting other molecules, such as LAG-3, TIM-3, and B7-H4 [77].

Encouraging results on the usefulness of PD-1/PD-L1 blockade to treat lymphomas were reported in different clinical trials in the last decade. For instance, the treatment of adult patients with relapsed/refractory cHL with the anti-PD-1 antibody nivolumab was well tolerated and allowed better overall response (NCT01592370, NCT02181738) [78,79]. Similar results with relapsed/refractory cHL were also observed with another anti-PD-1 drug, pembrolizumab, in a Phase II trial (NCT02453594) [80], and this ICI drug also provided a better overall response for adult patients with relapsed/refractory primary MBCL (NCT02576990). Nivolumab was the first ICI against PD-1 approved in the USA for use in hematological malignancies, approved for adult patients with relapsed/refractory cHL Pembrolizumab is approved for adult and pediatric patients with relapsed/refractory cHL, and for relapsed/refractory primary MBCL patients who relapse after two or more lines of primary therapy.

Despite the known or anticipated advantages of using ICIs to treat cancers that express a high level of its regulators, immune checkpoint blockade protocols are not free of limitations and caveats to be considered and better understood. For instance, the use of ICIs drugs can lead to an increased number of hyper-responsive T-cells in the periphery, resulting in inflammatory toxicities, known as immune-related adverse events (irAEs). Most irAEs are light to mild, although, in rare cases, can lead to death. Overall, they are well-tolerated and do not adversely affect the treatment [81]. Besides irAEs, one of the major characteristics that hampers a more successful use of ICIs is therapy resistance, essentially categorized as primary, adaptative, or acquired [82].

Another important point is that more immunogenic tumors usually have better responses to ICIs, and the composition and humoral profile of the tumor immune microenvironment (TIME) is a key factor affecting the success of immune checkpoint blockade strategies. The TIME can be classified as cold, when the tumor lacks immune cell infiltrates, or hot, on tumors highly inflamed, showing a high number of T-cells and high production of pro-inflammatory cytokines. Hot TIMEs tend to respond better to immunotherapy, including PD-1/PD-L1 blockade [83]. For instance, the TIME in cHL benefits immune checkpoint blockade therapy, as this lymphoma features extensive non-malignant immune infiltrate with CTLA-4+ T-cells [84], in addition to the very scarce neoplastic HRS cells that commonly express PD-Ls [85,86]. Conversely, BL consists of over 90% of malignant cells, with a very scarce TIME formed by some macrophages and rare T-cells. Furthermore, myc alterations in BL provide growth and survival signals to the neoplastic cells (which could explain the independence of a microenvironment for survival) [87], and they lack significant PD-L1 expression [85,88]. These features make it very challenging to design therapeutic protocols with immune checkpoint blockade for BL treatment, and there are no approved drugs for this disease so far.

The BL is poorly immunogenic, at least partially because the EBV-infected neoplastic cells express viral latency type I, very restrictive in terms of expression of viral products, notably immunogenic ones. Therefore, inducing a less restricted latency program (types II or III) could make the immunological cold TIME in BL more immunogenic and responsive to immune checkpoint blockade. Based on these premises, the BL cell lines KEM1, MUTU1 (RRID: CVCL_7202), and Rael (RRID: CVCL_7208) were treated with decitabine, a hypomethylating agent, which caused a significant increase in the activity of EBV Cp promoter, upregulating the expression of LMP1 and EBNA3C at protein levels and inducing a latency III. The decitabine treatment in xenografted mouse tumors in vivo caused EBNA2 expression in KEM1, MUTU1, and Rael -derived tumors, and LMP1 in xenografts of KEM1 and MUTU1, but not Rael. To test the durability of decitabine’s effects, LMP1 and Cp expression was analyzed up to seven days after drug washout, and results showed a minimal decrease in the expression of those genes. For Rael xenografts, EBNA2 was analyzed by IHC and results showed little to no decrease in EBNA2+ cells, with some areas of the tumor remaining EBNA2+ even after over 60 after washout [89]. As LMP1 and EBNA2 are known to modulate the expression of PD-L1 and other immune checkpoint regulators, the drug-induced expression of these proteins could potentially upregulate PD-L1 expression in lymphomas with low PD-L1 expression, making them more vulnerable to ICIs.

Cancer cells may also not respond to immune checkpoint blockade due to the absence of tumor antigens, or by mechanisms that impact the MHC-mediated antigen presentation, allowing the immune escape of the neoplastic cells. Loss or dysfunctions in genes encoding components of the antigen processing machinery, such as beta-2-microglobulin (B2M) or MHC class I molecules, have been shown to mediate checkpoint blockade resistance [83]. Mutations in the B2M gene or loss of MHC are found in different lymphomas, but they appear to be less frequent in those associated with EBV infection. In tumor biopsies of 111 DLBCL patients, B2M mutations were found in GC-type DLBCL and ABC/NOS subtypes, but not in BL samples [90]. Thus, it is an open question whether the resistance to immune checkpoint blockade due to loss/dysfunctions of antigen processing genes is eventually less frequent in EBV-positive lymphomas.

Cyclin-dependent kinases (CDK) 4 and 6 are important drivers of the cell cycle and are relevant to oncogenesis as they promote the progression of many cancers. In vivo studies of solid cancers have shown that inhibitors of CDK4/6 potentiate antigen presentation of tumor cells, increase expression of IFNγ of effector T-cells, and decrease expression of inhibitory immune checkpoints (e.g., PD-1, PD-L1, TIM-3) in CD8+ T-cells. In this context, treating tumors with CDK4/6 inhibitors greatly enhances the effects of anti-PD-1 therapy [91,92]. In a study published in 2009 [93], a potential role of B7-H4 was demonstrated in inducing growth-arrest of EBV+ lymphoblastoid cell lines in vitro, by downregulating gene expression of CDKs (including CDK4/6) and cyclins. Although better proof-of-concept data are necessary to conclude whether EBV-associated cancers may behave differently regarding immune checkpoint blockade response, these results provide a hint on whether cancers associated with EBV infection can achieve a more favorable response to ICIs.

In a clinical trial to assess the response to pembrolizumab in patients with metastatic gastric cancer, six out of the 61 patients enrolled had EBV-positive GaC tumors (9.8%). The overall response rates (ORR) to pembrolizumab in this group of patients was 100%, and 50% for the patients with PD-L1-positive tumors. Patients with high microsatellite instability (MSI) also had better responses, and PD-L1 positivity correlated with EBV-positivity and high MSI, even though EBV infection and MSI were mutually exclusive [94]. The response rate to pembrolizumab was also analyzed in a small cohort of patients with relapsed/refractory NHL, including tumor samples of DLBCL, NKTCL, peripheral MBCL, and TLBL. All samples of NKTCL (*n* = 14) and one out of four (25%) samples of peripheral MBCL were EBV-positive and showed higher PD-L1 expression compared to the EBV-negative tumors. Patients with EBV-positive tumors and high PD-L1 expression responded better to the therapy, especially those with NKTCL, while no response was observed for patients with relapsed/refractory EBV-negative NHL tumors in this cohort [95]. Some case reports also describe a good response to ICIs in rare cancers strongly associated with an EBV infection, as is the case of the primary lymphoepithelioma-like carcinoma (LELC) of the lungs [96,97].

Even though there are still scarce data regarding possible connections between the EBV and a more favorable response to immune checkpoint blockade therapies in cancers—either intrinsic or potentially induced by managing the viral infection—a new and promising perspective has emerged for the clinical management of cancers. Considering either tumor-associated or not with the EBV infection, and understanding the mechanisms by which the EBV-positive tumors behave differently in terms of ICIs susceptibility, can provide valuable insights to overcome therapy resistance in malignant neoplasms in general.

## 6. Conclusions

Major advances in the identification of immune checkpoints and their clinical use have been made lately, along with a glimpse of potential mechanisms that regulate their expression. Cancer therapies based on immune checkpoint blockade have been approved for the treatment of different relapsed/refractory lymphomas, and they may even become the first line of treatment for some diseases, including solid tumors. Different viruses have the potential to cause human cancers and understanding how they exploit the hosts’ defenses is strategic for the control of cancer progression and even the cure of virus-associated malignancies. Nonetheless, research on the EBV takes the lead once more, by showing viral properties important to better understand the tumor immunity and providing information that may be pivotal for the treatment of the EBV-associated cancers. Achieving a better understanding of the viral regulation of immune checkpoints can be invaluable also for cancer management in general, for instance, due to the identification of targetable immune checkpoint components and how to avoid or overcome cancer resistance to ICI-based therapies. This requires further investigation on the interaction of viral products and the expression of immune checkpoints regulators, along with experimental and clinical studies about the functional effects of the EBV modulation of immune checkpoints.

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
