# Peer review of "The Epstein-Barr Virus Hacks Immune Checkpoints: Evidence and Consequences for Lymphoproliferative Disorders and Cancers"

_biomolecules, 2022, doi:10.3390/biom12030397_

Round 1

Reviewer 1 Report

This review describes how EBV regulates PD-1/PD-L1 and impacts the response to checkpoint blockade, but there are major problems with it.

  1. It is not comprehensive. For immune checkpoints, only PD-1/PD-L1 is described in this review, and only lymphoma is included for EBV-associated cancers. This is not relevant to the title of the review.
  2. This review focuses on the mechanism of Epstein-Barr virus regulation of PD1/PD-L1, but it lacks innovation. There have been two other similar reviews published this year, and they summarized much more clearly and comprehensive, especially they used figures to clarify these aspects. These references are: Murata T, et al., Microorganisms.; Li X et al., International Immunopharmacology. There is no figure, which would help for clearer description.
  3. There are too many details that not closely relevant to the topic of the manuscript. In the section of introduction, the paper introduces the general situation of cancer incidence, the types of cancer caused by EBV and the types of infection of EBV, which are not closely related to the main topic and should be summarized briefly. By the way, the latest edition of Global Cancer Statistics has been published in 2020. The authors only refer to the 2018 edition in the article, and the corresponding data should be updated. The types of lymphoma and Hodgkin's lymphoma are also described too long.
  4. The logic needs to be improved. In lines 347-365, about the prognosis, which can be put together with “Clinical impact of immune checkpoints in human beings”. And, the clinical impact should be put at the end of the article, in front of the conclusion.

Author Response

Firstly, we would like to thank the reviewer for the careful evaluation of the manuscript and the constructive criticism made!

Please find below our reply (R) to each of the specific comments provided:

This review describes how EBV regulates PD-1/PD-L1 and impacts the response to checkpoint blockade, but there are major problems with it.

  1. It is not comprehensive. For immune checkpoints, only PD-1/PD-L1 is described in this review, and only lymphoma is included for EBV-associated cancers. This is not relevant to the title of the review.
    R:  We agree that the original title was broad and did not properly address the focus of this review. Based on your comment, we also aimed to make it more tempting; thus, we changed the title to “Epstein-Barr virus hack Immune Checkpoints: Evidence and consequences for lymphoproliferative disorders and cancers”.
    Regarding the review’s coverage, we intended to make the review more concise considering that the Biomolecules Journal has a broad audience, not targeting specifically virologists, immunologists, and cancer biologists. Therefore, our eview aimed to highlight the published evidence that allows the reader to understand general mechanisms by which EBV can regulate immune checkpoints and what are the documented or expected consequences of it for EBV-associated cancers, with focus on lymphomas. Nonetheless, we provided information for other important EBV-associated cancers, such as nasopharyngeal and gastric carcinomas.
    Regarding PD-1/PD-L1, these molecules are mentioned more frequently essentially due to a literature bias -- we now added a statement about it in the revised manuscript (section 3, lines 167-174). We did seek to provide relevant information for other immune checkpoints molecules, such as B7-H4, LAG-3, PD-L2, and TIM-3.
    We hope that these clarifications make sense to the reviewer, but we will be glad to make additional changes if the reviewer understands that are essential to make this manuscript a more valuable contribution, considering its scope and targeted audience.

  2. This review focuses on the mechanism of Epstein-Barr virus regulation of PD1/PD-L1, but it lacks innovation. There have been two other similar reviews published this year, and they summarized much more clearly and comprehensive, especially they used figures to clarify these aspects. These references are: Murata T, et al., Microorganisms.; Li X et al., International Immunopharmacology. There is no figure, which would help for clearer description.
    R: Although the indicated articles have some similarities with our review, we understand that they are quite different and do not overlap in key aspects. The paper by Murata and colleagues focuses on the role of herpesvirus in general in the regulation of PD-1/PD-L1 molecules only, with EBV being briefly summarized. Li and colleagues more densely discuss EBV-associated cancers and its regulation of PD1/PD-L1, but it focused on a proposed potential role of PGE2 in EBV-induced immunosuppression, and its potential relationship to PD-L1. Both papers did not address the available data on EBV’s regulation of other immune checkpoints, what we do -- even considering that the studies available are still scarce. Furthermore, our review stands out from those papers as it proposes how the management of EBV-associated cancers may benefit from the current knowledge on how EBV exploits the immune checkpoints. Lastly, we explicit  the important gap on the research of EBV’s regulation of other important immune checkpoint molecules, aiming to call the attention of the scientific community to this important research opportunity that is been overlooked.
    Regarding the figure, because the Graphical Abstract we provided was not mentioned, it is unclear to us whether the reviewer proposes that we include another one, or our figure actually was overlooked for some reason.

  3. There are too many details that not closely relevant to the topic of the manuscript. In the section of introduction, the paper introduces the general situation of cancer incidence, the types of cancer caused by EBV and the types of infection of EBV, which are not closely related to the main topic and should be summarized briefly. By the way, the latest edition of Global Cancer Statistics has been published in 2020. The authors only refer to the 2018 edition in the article, and the corresponding data should be updated. The types of lymphoma and Hodgkin's lymphoma are also described too long.
    R: To address this comment, we edited the Introduction section substantially aiming to make it more concise and straightforward. Nonetheless, we would like to keep some information because of our understanding that the audience of Biomolecules Journal is broad and heterogeneous, as mentioned in our reply to comment #1. This is the case of essential information about the lymphomas, which are not common knowledge for researchers not working with these diseases.  Finally, we updated the reference presented - we are grateful to the reviewer for noticing it!

As a final word, we deeply appreciated the reviewer’s comments, which allowed us to improve and update further the manuscript with newly published studies, including a recent review in Biomolecules (MÜNZ C. Biomolecules 12:38, 2021. DOI: 10/gpg82z) that is complementary and serendipitously well-articulated with our own.

Reviewer 2 Report

The Authors wrote a review about "Epstein-Barr Virus regulation of Immune Checkpoints in Human Cancers." The Epstein-Barr virus is one of the most common viruses in people and at the same time the mysterious cause of many diseases, including cancer. The topic on which the authors undertook to write is still relevant, especially in the aspect of searching for effective therapies for the treatment of cancers related to this oncogenic virus.

The review is interesting but has some deficiences that need to be filled.

Major revision:

  1. Line 103-113. After reading the manuscript, it lacks references to the regulation of immune checkpoints in other EBV-related cancers than lymphoid origin (EBV-associated lymphomas). In the chapter „Introduction”, Authors will explain their choice, which I partially accept. Please enter in chapter „Introduction” or chapter 5. „EBV-associated cancers and resistance to immune checkpoint blockade” references regarding the regulation of immune checkpoints in other cancer associated with EBV. Such information will greatly enrich the manuscript and will be a valuable source of information and citations for other researchers. The information given in chapter 5 is too laconic and leaves us unsatisfied.
  2. Please consider changing the manuscript title to a more precise "Epstein-Barr Virus regulation of Immune Checkpoints in chosen Human Cancers"
  3. Why in the manuscript only the biological role of PD-1 and CTLA-4 (Line 192-197) was described. No description for LAG-3, TIM-3, VISTA. Please complete. Add this information in chapter "Clinical impact of immune checkpoints in human cancers" (Line 375).

Minor Revision:

Line 36: space before [2].

Line 42: space before [4].

Line 320: Cristino and colleagues (2019), should be Cristino et al. [62].

Line 335: Chakravorty and colleagues (2019), should be Chakravorty et al. [64].

Line 381, 383 : insert space

Line 395: insert space before „use”

Line 469: „In a study published in 2009, Park and colleagues …”, should be „In a study published in 2009 [89]…”

Author Response

Firstly, we would like to thank the reviewer for the careful evaluation of the manuscript and the constructive criticism made!

Please find below our reply (R) to each of the specific comments provided:

The Authors wrote a review about "Epstein-Barr Virus regulation of Immune Checkpoints in Human Cancers." The Epstein-Barr virus is one of the most common viruses in people and at the same time the mysterious cause of many diseases, including cancer. The topic on which the authors undertook to write is still relevant, especially in the aspect of searching for effective therapies for the treatment of cancers related to this oncogenic virus.

The review is interesting but has some deficiences that need to be filled.

Major revision:

  1. Line 103-113. After reading the manuscript, it lacks references to the regulation of immune checkpoints in other EBV-related cancers than lymphoid origin (EBV-associated lymphomas). In the chapter „Introduction”, Authors will explain their choice, which I partially accept. Please enter in chapter „Introduction” or chapter 5. „EBV-associated cancers and resistance to immune checkpoint blockade” references regarding the regulation of immune checkpoints in other cancer associated with EBV. Such information will greatly enrich the manuscript and will be a valuable source of information and citations for other researchers. The information given in chapter 5 is too laconic and leaves us unsatisfied.
    R: We revised and restructured the whole manuscript to address the reviewers' comments. Even though the main focus of the review are EBV-associated lymphomas, please refer to information that we also provided about the EBV-associated epithelial cancers nasopharyngeal and gastric, notably in section 3 (item "EBV regulation of immune checkpoints: Insights based on PD-1/PD-Ls upregulation") and section 4. Regarding section 5, it was reviewed and reorganized; we hope that the changes made are considered appropriate. Nonetheless, we will be glad to make aditional changes if the reviewer consider necessary to further improve the quality and value of this review.
  1. 2. Please consider changing the manuscript title to a more precise "Epstein-Barr Virus regulation of Immune Checkpoints in chosen Human Cancers"
    R: Thank you for stimulating the changes in the title. It was modified to "Epstein-Barr virus hack Immune Checkpoints: Evidence and consequences for lymphoproliferative disorders and cancers"

  2. Why in the manuscript only the biological role of PD-1 and CTLA-4 (Line 192-197) was described. No description for LAG-3, TIM-3, VISTA. Please complete. Add this information in chapter "Clinical impact of immune checkpoints in human cancers" (Line 375).
    R: Originally only the biological roles of PD-1/PD-Ls and CTLA-4 were covered because these molecules were more investigated and their importance to immunotherapy. Nonethless, as per the reviewer suggestion, we now briefly present information about LAG-3, TIM-3, and VISTA. Please refer  to section 4, lines 334-341.

Minor Revision:

  • Line 36: space before [2].
    R. Done
  • Line 42: space before [4].
    R. Done
  • Line 320: Cristino and colleagues (2019), should be Cristino et al. [62].
    R. The text were changed ommiting authors' names, as follows, : "Considering EBV-driven mechanisms of PD-Ls upregulation, the 3’-UTRs of both PD-L1 and PD-L2 were found to be targets by the viral miR-BHRF1-2-5p, as experimentally confirmed by luciferase reporter assays. Lymphoblastoid cell lines (LCL) infected with the EBV strains WIL (EBV-WIL) or B95-8 (EBV-B95-8) showed a significant reduction in PD-L1 and PD-L2 when treated with miR-BHRF1-2, indicating that this viral miR-RNA regulates the PD-Ls [60]." 
  • Line 335: Chakravorty and colleagues (2019), should be Chakravorty et al. [64].
    R. Done
  • Line 381, 383 : insert space
    R. Done
  • Line 395: insert space before „use”
    R. Done
  • Line 469: „In a study published in 2009, Park and colleagues …”, should be „In a study published in 2009 [89]…”
    R. Done

We greatly appreciate the reviewer's attention and contributions for improving this manuscript, thank you!

Round 2

Reviewer 1 Report

The authors have made sufficient modifications according to the comments excepting several grammar errors, such as line 63,108,122. Please check the whole manuscript carefully. Regarding the figure, the authors mentioned the Graphical Abstract, but I cannot find it in the manuscript or supplementary files. Please check and insert it.

Author Response

Dear Reviewer #2,

We apologize for the English problems found, at the same time that we are sincerely grateful for your careful inspection of the text. We revised the manuscript again, double-checking for grammar and other language issues, including those indicated in your comment. 

We also feel sorry to hear that the Figure (Graphical Abstract) that we uploaded in the previous submissions was not available for your analysis. We could not identify the problem; nonetheless, to be sure that it will be available in this new submission, we requested the Journal's Editorial Office to include the file in the system. Accordingly, the graphical abstract should be available as supplementary material for your analysis, but we are also providing it as a PDF file attached to this reply (the system renamed it to author-coverletter-17933548.v1.pdf).

Thank you very much for your valuable comments and support to improve the manuscript!
